

# Changes in the Si/P weathering ratio and their effect on the selection of coccolithophores and diatoms.

Virginia García-Bernal[1], Óscar Paz[1], Pedro Cermeño[1]

[1]Department of Marine Biology and Oceanography, Institute of Marine Sciences (ICM-CSIC), Passeig Marítim de la Barceloneta, 37–49, Barcelona, 08003, Spain

*Correspondence to*: Pedro Cermeño (pedrocermeno@icm.csic.es)

**Abstract.** The relative contribution of coccolithophores and diatoms to export production and burial influences the mechanisms of carbon (C) removal from Earth's surface reservoirs. Whereas the supply of phosphorus (P) to the ocean basins controls export production over geological timescales, the availability of dissolved silica (Si) determines the relative contribution of diatoms to export production and increases the efficiency of the ocean's biologically-driven C sequestration. Here we combine analyses of data from the sedimentary record and Earth system model simulations to investigate the coupling of P and Si weathering fluxes over the last 40 million years (My). Although both weathering fluxes increased on average through time, the relative increase of Si weathering over the last 20 My exceeded that of P. Our analysis suggests that the increase in P weathering under a regime of increasingly ocean turbulence allowed full utilization of increased Si supplies and links the ecological success of marine diatoms, and consequential decline of coccolithophores, to changes in the Si/P weathering ratio. These results allow us to envisage that ocean fertilization initiatives would be more effective if they were integrated within the framework of geo-engineering programs aimed at enhancing silicate weathering.

**Keywords:** diatoms, coccolithophores, continental weathering, nutrient supply ratio, sedimentary record, biogeochemical modelling

## 1 Introduction

Massive deposits of chalk, up to a thousand meters thick, in many parts of the world bear firm testament to the great supremacy of calcareous nannoplankton, chiefly coccolithophores, in Cretaceous and early Cenozoic oceans (Bown et al., 2004). The long supremacy of coccolithophores is eventually usurped during the second half of the Cenozoic era by siliceous phytoplankton, chiefly diatoms, which continue their ecological ascension and leading role in marine plankton ecosystems until the present (Falkowski et al., 2004;Katz et al., 2004).

Diatoms contribute up to 40% of modern ocean primary production, thus playing an important role in the carbon (C) cycle (Treguer et al., 1995;Smetacek, 1999). Their absolute requirement for silicic acid (Si), in the form of orthosilicic acid, which they use to construct the cell wall, influences their oceanic distributions, and constitutes a critical determinant of their



exceptional fossil record in marine sediments (Cortese et al., 2004;Armbrust, 2009). Diatom dominate phytoplankton community biomass in temperate and polar seas, and coastal environments, where nutrient-rich deep waters episodically enter the euphotic zone (Margalef, 1978). When diatom blooms collapse, they sink rapidly into the deep ocean increasing the concentration of silica and C in the bottom waters and sediments (Tréguer and De La Rocha, 2013;Cermeño et al.,

2015;Smetacek, 1999). This is why diatoms are considered to be important players in the cycling of silica and C (Cortese et al., 2004;Falkowski et al., 2004).

Coccolithophores also play an important role in the C cycle owing to their ability to produce ornamented plates of calcium carbonate ($CaCO_3$), known as coccoliths. The calcification process decreases the alkalinity of seawater and releases $CO_2$, which may escape to the atmosphere. Furthermore, a decrease in the $CaCO_3$ to organic C rain ratio increases the

solubility of deep sea sedimentary calcite, which enhances the ocean's storage capacity for atmospheric $CO_2$. Hence, the relative contribution of coccolithophores and diatoms to marine export production potentially influences the $CaCO_3$ to organic C rain ratio, a major factor determining the partitioning of C between the atmosphere and ocean (Cermeño et al., 2008).

The different approaches used by biologists (seawater samples) and paleontologists (fossil records) to study the

ecology of coccolithophores have provided two opposing views concerning their distribution patterns in the ocean. According to biologists (Balch, 2004), the coccolithophores are primarily found in stratified environments such as the subtropical ocean gyres, and oligotrophic seas such as the Mediterranean (Balch, 2004;Margalef, 1978). Conversely, the paleontologists (Geological School according to Balch, 2004) contend that coccolithophores can dominate in mixed environments of high productivity typical of, for instance, coastal seas and upwelling systems (Cachão and Moita, 2000).

These rather distinct views might simply reflect differences in the rates of preservation of sedimentary $CaCO_3$ and silica (Balch, 2004), and the particular features of certain coccolithophore species such as *Emiliania huxleyii* that possesses physiological traits characteristic of fast growing species.

Resource competition theories provide a basis to account for the fortunes of coccolithophores and diatoms in the past. The classical ecological succession from non-siliceous plankton to increased importance of diatoms is explained as a

result of changes in the ratio between the supply of Si and other nutrients such as phosphorus (P) (the 'resource ratio theory') (Fig. 1a). The resource ratio theory predicts the replacement of species as a function of the ratio of limiting nutrients, and has been well corroborated under laboratory controlled conditions (Egge and Aksnes, 1992;Grover, 1997;Tilman, 1982). Alternatively, the 'dynamical nutrient supply theory' postulates that the timing of delivery of nutrients provides an ecological selection pressure (Grover, 1991;Margalef, 1978) (Fig.1b). Based on the conceptual foundations of the 'mandala',

resource-based competition models suggested that intensification of upper ocean turbulence and nutrient supply dynamics would potentially increase the ecological success of fast growing diatoms ("r-strategist") at the expense of the slower-growing coccolithophores ("K-strategists") (Tozzi et al., 2004;Cermeño et al., 2008;Cermeño et al., 2011).

The resource ratio and the dynamical nutrient supply theories constitute major selective pressures (Falkowski and Oliver, 2007;Kooistra et al., 2007). The former is dictated largely by continental weathering, which controls the fluxes of Si




and other nutrients to the ocean basins on time scales of millions of years. The latter is dictated by upper ocean turbulence and hence the temperature gradients between the equator and the poles, which can be forced on time scales ranging from decades to millions of years. Sedimentary evidence indicates that continental weathering fluxes and upper ocean turbulence increased concurrently during the second half of the Cenozoic era (Zachos et al., 1999;Anderson and Delaney, 2005;Isaacs and Petersen, 1987;Berger, 2007;Wright, 2001). However, specific details such as the coupling of P and Si weathering fluxes and its relation to changing ocean turbulence are essential to substantiate the prediction of resource competition theories about the distinct fortunes of coccolithophores and diatoms in the geological past .

The main objective of this research was to investigate the linkage between the Si/P weathering ratio and the ecological selection of coccolithophores and diatoms over the last 40 million years (My). To address this question, we used two complementary approaches. First, we analyzed sedimentary records of P burial rate and lithium (Li) isotopes, which have been previously used to infer continental weathering fluxes of P and Si, respectively (Follmi, 1995;Cermeño et al., 2015). Second, our sediment–based data analysis was substantiated by an Earth system biogeochemical model (COPSE) (Bergman et al., 2004), which allows P and Si weathering fluxes to be simulated through time. The resulting P and Si weathering fluxes estimates were then compared with the fossil record of coccolithophores and diatoms.

## 2 Material and methods

### 2.1 The sedimentary record: P and Si weathering fluxes

Figure 2 shows the geographic locations of samples from which P burial rates and Li isotopes used to calculate P and Si weathering fluxes were obtained (Follmi, 1995;Misra and Froelich, 2012). Integrated over representative areas and over time scales longer than the residence time of P in the ocean, variations in marine P burial rates are a good approximation of variations in the weathering flux of P into the marine environment. The bioavailable P pool —the input of dissolved P plus the particulate P which is converted to dissolved fraction— is often referred to as reactive P (Froelich et al., 1982;Compton et al., 2000). Estimates of reactive P flux to the marine environment, based on the analysis of pristine rivers, are in the range of 0.1-0.3 Tmol·y$^{-1}$. Thus, we calculated the global P weathering flux over the last 40 My using the normalized to present biogenic P burial curve reported in (Follmi, 1995) and modern global flux estimates  (Fig. 3a).

The weathering of continental silicates is the main source of dissolved Si to the ocean basins. We used estimates of dissolved Si flux based on the seawater lithium isotope record ($\delta^7$Li) (Misra and Froelich, 2012) as described in (Cermeño et al., 2015). They first calculated the flux of suspended Si to the ocean basins by assuming that the global average ratio of suspended to dissolved Si in modern rivers is 4:1, while at 60 My it was 1:4. This change in the ratio of suspended to dissolved load in rivers is derived from a global lithium mass balance (Misra and Froelich, 2012), that accounts for the fluxes to and from the ocean basins, and reflects the increasingly incomplete weathering of continental silicates driving a significant increase in particulate fraction. Li and Elderfield calculated an approximately two-fold increase in dissolved Si flux over the same time period (Li and Elderfield, 2013). Hence, Cermeño et al imposed a twofold linear increase in dissolved Si content





of riverine input over the last 50 My, calculated the flux of suspended Si to the ocean, and modified the dissolved Si flux accordingly (Fig. 3b) (Cermeño et al., 2015). The resulting normalized to present curve was multiplied by the global estimate of dissolved Si output modern river of 5.8 Tmol·y[1] (Tréguer and De La Rocha, 2013).

The Summed Common Species Occurrence Rate (SCOR) of coccolithophores and diatoms was used as a proxy of plankton functional group dominance (Hannisdal et al., 2012;Cermeño et al., 2015). The SCOR is based on the assumption that the more globally abundant a species is, the more likely it is to occur in a greater number of sampling sites. For each individual species the probability of detection is calculated as,

$$\lambda = \ln(1 - p_{ij}), \tag{1}$$

with $p_{ij}$ estimated as $y_{ij}/n_j$, where $y_{ij}$ is the number of samples in which species $i$ is present at time $j$ and $n_j$ is the number of available sites in time bin $j$. Available sites are those that contain at least one species in the set of common species included in the analysis. Then, the SCOR is computed as the total density of a given set of $m_j$ species in a particular time bin as,

$$SCOR_j = \text{sum } [m_j * \lambda_{ij}]. \tag{2}$$

As $p$ approaches 1, the rate of increase in $\lambda$ grows rapidly, such that common/widespread species have a much greater influence on SCOR than less common species. The value of this index is founded on the idea that most fossil species have a hat-shaped temporal occurrence trajectory, such that individual species increase their geographic ranges gradually through time, stay at their maximum for a relatively short period of time, and then gradually decline to extinction (Liow et al., 2010). Thus, the larger number of species at their characteristic maximum occupancy, the greater the SCOR.

## 2.2 Earth system biogeochemical modelling

We performed numerical simulations within the framework of COPSE (Carbon-Oxygen-Phosphorus-Sulfur-Evolution), an Earth system biogeochemical model that couples interactive and evolving terrestrial and marine biota to geochemical and tectonic processes. Among other relevant quantities, COPSE computes the continental weathering fluxes of P and Si to the ocean basins, being these fluxes and their coupling the main objective of this research.

Total P weathering is computed from the estimated weathering fluxes of continental silicates and carbonates, and from the concurrent oxidation of sedimentary organic matter. The relative proportion of P in each rock type is a priori prescribed in the model. Reactive P is delivered to the ocean via river runoff, consumed by organisms in the sea and eventually buried in marine sediments. Organic P is rapidly remineralised by bacterial assemblages returning inorganic P to the oceanic reservoir. Only a minor fraction of P becomes sequestered in the sediments over geologic time scales. On time scales longer than the residence time of P in the ocean (20-100 ky), this P burial rate serves as an approximation to the reactive P weathering flux. COPSE computes marine organic C burial as the square root of new primary production. Then, the marine organic P burial is obtained by applying a C:P ratio of roughly 250:1. This number is a bit higher than the usual Redfield ratio of 106:1, albeit a more elaborated model based on the ocean anoxic fraction (VanCappellen and Ingall, 1996) is also available.



The flux of dissolved Si to the ocean basins is assumed to be proportional to the chemical weathering rate of continental silicates. Most Earth system biogeochemical models assume that the main control on silicate weathering is volcanic degassing, which increases atmospheric $CO_2$ levels and global temperature (Berner et al., 1983;Bergman et al., 2004). Silicate weathering is accordingly coded using the flux function:

5          $silw \propto U \cdot f\,(CO_2),$                                                                (3)

where $U$ represents uplift (an external forcing), and $f(CO_2)$ is the silicate weathering dependence on atmospheric $CO_2$ concentration. This function is inherited from GEOCARB (Berner, 1991, 1994;Berner and Kothavala, 2001;Berner, 2006), a previous Earth system model for the geologic evolution of the C cycle. In its present configuration the model predicts a steady decrease in silicate weathering from ~40 million years ago (Ma) to present linked to a concomitant reduction of

volcanic degassing.  This result is somewhat surprising taking into account that strontium (Sr) and lithium (Li) isotope ratios suggest a remarkable increase in continental weathering largely attributed to the uplift and thrusting of the Himalayas (Raymo&Ruddiman 1992, Misra&Froelich 2012). This apparent discrepancy results from the fact that the prescribed flux function is heavily dependent on atmospheric $CO_2$ concentrations. The full expression for *silw* indeed involves the following terms: carbonate C degassing, amount of carbonate C, uplift rate, weathering dependence on atmospheric $CO_2$ concentration

and the effect of land biota on weathering. In the model's default configuration, from 40 Ma to present, $f(CO_2)$ accounts for a reduction in silicate weathering by a factor of ~2.5, whereas $U$ accounts for an increase in silicate weathering by a factor of ~1.5 within the same time span. Since *silw* is basically modelled as the product of these two variables, the final outcome is a steady decreasing silicate weathering curve (Bergman et al., 2004).

Recently, Maher & Chamberlain have pointed out that silicate weathering might be more related to runoff and

topography (Maher and Chamberlain, 2014). The riverine flux of weathered minerals is expected to be dependent on the balance between the time that minerals are exposed to fluids and their dissolution kinetics. Maher & Chamberlain's model largely builds upon the concept underlying the Damköhler coefficient, $Dw$ (m y$^{-1}$), which can be defined in a simplified form as:

          $Dw \sim \lambda\,/\,\tau,$                                                                            (4)

where $\lambda$ is an effective flow-path length (related to orography, river drain network, rock porosity, etc) and $\tau$ is a characteristic equilibrium time (related to solute concentration, mineral solubility, reaction rate, etc). $Dw$ compares how long it takes a dissolution reaction to reach its thermodynamic limit against the flow-path length for the solutes to reach the ocean basins. In other words, for a fixed runoff velocity: the faster the reaction rate, the higher the delivery of solutes; or alternatively, the longer the flow-path length, the larger the total amount of mineral that can be dragged. Maher & Chamberlain relate this

number to the cratonic/collisional nature of the physical environment, setting a value of 0.03 m y$^{-1}$ as the frontier between both regimes (Maher and Chamberlain, 2014). Since COPSE does not handle the runoff variable or any of the parameters involved in the calculation of $Dw$, our approach consisted on finding an approximation for the weathering flux-runoff dependence following Maher and Chamberlain's empirical relationships between runoff and $Dw$ (Maher and Chamberlain, 2014). Ultimately, our objective was to construct a **weathering flux-uplift relationship** to be implemented into the COPSE





model. Therefore, we started from the 3d-surface (runoff, $Dw$, flux) reported by Maher & Chamberlain, then fitted the available rivers data to a linear regression model and projected it onto the ($Dw$, flux) plane, to finally apply a matching criterion upon $Dw\leftrightarrow$uplift variables by assuming a direct correspondence between the min/max values of both magnitudes.

# 3 Results

We compared estimates of P and Si weathering flux through time using data from the sedimentary record and model simulations (Fig. 3). As for the model simulations we tried up 6 different parameterizations, based on Maher and Chamberlain' description of silicate weathering; i.e., six different combinations of the min/max Damköhler coefficient number, aimed at testing the sensitivity of the model outcomes to changes in the prescribed model function (weathering flux-uplift relationship). For each nutrient, all simulations exhibited similar patterns. P weathering fluxes increased across the

Oligocene and mid-Miocene, and decreased in the late Eocene, early Miocene and late Miocene (Fig. 3a). Our model simulations were remarkably coincident with the P weathering curve derived from the analysis of the sedimentary record (Fig. 3a). Si weathering fluxes showed a slight increase ~40 Ma, followed by huge increase over the last 20 My (Fig. 3b) . Similarly, the model simulations fitted satisfactorily the Si weathering flux curve arising from the analysis of the sedimentary record (Fig. 3b) (Cermeño et al., 2015). Among the 6 different model outcomes, the best fit was selected after

least squares analysis to the empirical sedimentary data, resulting $z1022$ as the closest model for both P and Si.

Our theoretical rationale was aimed at linking relative changes in the P and Si weathering fluxes (i.e., Si/P weathering ratio) to the selection of coccolithophores and diatoms (Fig. 1a). The Si/P weathering ratio was calculated dividing the weathering fluxes through time for each nutrient. The precise value of the weathering ratio was dependent on estimates of modern river fluxes of reactive P and dissolved Si, and their associated uncertainties. Overall, P and Si

weathering fluxes increased over the past 40 My (Follmi, 1995;Cermeño et al., 2015). The Si/P weathering ratio increased primarily across the Eocene/Oligocene (E/O) (~35-31 Ma), early Miocene (~23-17 Ma) and late Miocene (~12-5 Ma), and remained relatively high until the present (Fig. 4a). We found that changes in the Si/P weathering ratio were consistent with peak diatom SCOR across the E/O boundary and the late Miocene (Fig. 4b). The diatom to coccolithophore SCOR ratio increased steadily since the early Miocene to the Pliocene and decreased slightly onwards (Fig. 4c). From the E/O boundary

to the early Miocene the Si/P weathering ratio decreased substantially (Fig. 4a). This trend was paralleled by increased P supplies and relatively unaffected Si fluxes, which provides a feasible explanation for the diatom crash during the Oligocene (Fig. 4b), when diatom SCOR returned to late Eocene values.

Overall, the patterns depicted by the SCOR of diatoms and coccolithophores were consistent with changes in the Si/P weathering ratio (Fig. 4). The 'resource ratio theory' predicts that an increase in the Si/P weathering ratio will select

diatoms at the expense of coccolithophores. However, the availability of Si is not a sufficient condition for the selection of diatoms, which require dynamical nutrient supplies to be competitively superior (Fig. 1b). Thus, to account for the rise of marine diatoms to ecological prominence, the Si/P weathering ratio and the effect of ocean turbulence on nutrient supply



dynamics, i.e., the 'dynamical nutrient supply theory', must be considered concurrently. The latitudinal sea surface temperature gradient of the past is recorded in the $\delta^{18}O$ signature of the calcium carbonate tests of planktonic and benthic foraminifera preserved in the sedimentary record. This is because benthic foraminifera are influenced by bottom waters, which typically form at high latitudes, whereas planktonic foraminifera are influenced by surface waters. The $\delta^{18}O$ signals

5 from 40 Ma to present suggest a continuous increase in the latitudinal temperature gradient, which intensified global wind stress and upper ocean turbulence in temperate and polar seas and upwelling systems (Wright, 2001;Falkowski and Oliver, 2007). Together the marine sedimentary record and our model simulations support the idea that increased P weathering under a regime of increasingly turbulent ocean conditions allowed full utilization of increased Si supplies.

## 4 Discussion

10 The last 40 My of Earth history have been dominated by striking changes in plate tectonics, global climate, ocean circulation and land biota (Molnar and England, 1990;Zachos et al., 2001;Kennett, 1977;DeConto and Pollard, 2003), with profound consequences on continental weathering fluxes, seawater chemistry and the ecology of marine phytoplankton functional groups such as coccolithophores and diatoms.

The main supply of dissolved Si to the ocean basins comes from the chemical weathering of continental silicates 15 (Holland, 1984). Throughout the Cenozoic, orogeny, marine regressions and the expansion of grassland ecosystems have been postulated as important factors contributing to the increase of dissolved Si fluxes to marine environments (Huh and Edmond, 1999;Falkowski et al., 2004). It has been hypothesized that the rise of marine diatoms during the Cenozoic era was related to i) superior competitive abilities for dissolved Si with respect to other siliceous organisms such as radiolarians (heterotrophic silicifiers) (Harper and Knoll, 1975) and ii) massive weathering of continental silicates in response to the 20 uplift of the Himalayas (Cermeño et al., 2015).

High topographic relief, steep slopes and rainfall facilitate the removal of soils and maintain mineral surfaces continually exposed to chemical weathering (Gaillardet et al., 1999;Millot et al., 2002). These physical environments are dominated by weathering-limited regimes in which the rate of bedrock supply exceeds the ability of the system to weather it and hence the flux of nutrients to the ocean is high. Weathering-limited regimes today are characterized by rivers draining 25 the Himalayas, the Tibetan Plateau and the Altiplano, with high monsoon and rainfall. Although Himalayan orogeny begins at ~50 Ma, crustal-scale thrusting was delayed for perhaps 20-40 My following the onset of collision (Yin and Harrison, 2000), and so did its impact on the chemical delivery to the ocean basins. A more direct link between Himalayan orogeny and the rise of diatoms to ecological prominence in marine ecosystems could be effective during the Miocene with the thrusting of the Higher Himalayan Crystalline sequences and the enhancement of monsoonal regimes (Cermeño et al., 2015).

30 Glacial activity can increase chemical weathering through two different mechanisms. First, glacial erosion is known to be capable of grinding solid bedrock and induce high chemical weathering rates (Prestrud Anderson et al., 1997;Montross et al., 2013). Second, ice growth decreases sea level, exposing large extensions of sedimentary deposits to subaerial



weathering. The massive appearance of continental glaciers and ice covers in the early Oligocene and the development of permanent ice-sheets on east Antarctica during the middle/late Miocene remobilized vast extensions of soils and regolith accreted over Antarctica prior to the onset of glaciation (Zachos et al., 1999;Thomson et al., 2013) and, presumably, increased weathering fluxes. These processes in conjunction with global increases in wind patterns and ocean turbulence

provided ecological advantages to marine diatoms (Falkowski et al., 2004;Margalef, 1978). These climatic conditions are representative of the E/O boundary (~36-33 Ma) and the cold climates of the Oligocene (34-23 Ma) and early Miocene (23 Ma). Indeed, previous evidence shows that, during this time, calcareous biogenic sediments were displaced northwards from the southern ocean by siliceous deposits (Kennett, 1977).

Land plants, and especially grasses, also contributed to accelerate the weathering of continental silicates. Up to 15%
of the dry weight of grasses consists of opal phytoliths being these micromineral deposits highly rich in particulate Si (amorphous silica). Grasses remained sparse until the E/O boundary (Retallack, 2001). During the mid to late Miocene, global cooling and aridification on the continents led to the expansion of grassland ecosystems, which provided a new source of bioavailable Si to the marine environment (Retallack, 2001;Strömberg, 2011). It has been hypothesized that the expansion of grassland ecosystems could boost the expansion of diatoms (Falkowski et al., 2004). However, empirical evidence to
substantiate this causal link is currently lacking.

Ultimately, the influence of mountain building and glacial activity on the Si/P weathering ratio depends on the extent to which these geological and climatic processes preferentially enhanced the dissolution of silicates or P-rich deposits. Sea level fall and subsequent exposure of sedimentary deposits to subaerial weathering is thought to be the main cause for increased P weathering fluxes to the ocean basins during the second half of the Cenozoic era (Follmi, 1995). Conversely,
long-term massive erosion of continental silicates caused by intense crustal deformation and relief formation in the Himalayas increased the flux of dissolved Si to marine environments. Thus, although sea level fall increased the flux of P to the marine environment, our analysis suggests that mountain building and glacial erosion elevated the Si/P weathering ratio. This idea is consistent with the observed decrease in the Si/P weathering ratio during the Oligocene (~30-23 Ma) and middle Miocene (~17-14 Ma) associated with massive ice-growth in Antarctica, which decreased sea level and limited weathering
due to extensive ice sheets over Antarctica.

Basalts weather relatively fast compared to other rocks (i.e. 2-5 orders of magnitude faster than quartz). The rapid extrusion of large igneous provinces increased the chemical delivery of nutrients. Apatite saturation in felsic magmas limits the P concentration in granitic rocks. P concentrations can be much higher in basalts, which in turn have lower Si contents (wt. <50%). The abrupt emplacement of the Ethiopian and Columbia River Flood Basalt provinces in the early Oligocene
(~30 Mya) and middle Miocene (~17-14 Ma), respectively, could have boosted basalt weathering. Thus, it is tempting to relate the observed decrease in the Si/P weathering ratio during the Oligocene and middle Miocene to these eruptive events.

The distinct fortunes of coccolithophores and diatoms during the Cenozoic era cannot be attributed exclusively to changing weathering fluxes and nutrient ratios. According to Margalef (1978), the combination of turbulence or variance in the components of vertical velocities, and the simultaneous supply of nutrients to the euphotic zone are considered important





factors in the biology and selection of marine diatoms and coccolithophores. Regardless of the concentration of dissolved Si in seawater, steady-state nutrient supplies to the euphotic zone typical of subtropical systems today, lead to prolonged nutrient limitation, which benefits phytoplankton with high nutrient affinity such as coccolithophores (Litchman and Klausmeier, 2008;Cermeño et al., 2011). High nutrient uptake rates and ability to store nutrients provide diatoms superior

competitive abilities in turbulent environments, where nutrients episodically enter the euphotic zone (Cermeño et al., 2008;Margalef, 1978;Cermeño et al., 2011). These environmental conditions begin to be prominent in the late Eocene when the onset of polar glaciation intensified the latitudinal thermal gradient (Wright, 2001), enhancing global winds, mesoscale activity and upwelling. This climatic transition has been associated with i) tectonically-driven changes in ocean circulation, which isolated Antarctica (Kennett, 1977) and ii) the decline of atmospheric $CO_2$ levels (DeConto and Pollard, 2003). In

summary, increased weathering fluxes, high Si/P nutrient supply ratios and concurrent enhancement of upper ocean turbulence and nutrient supply dynamics during the second half of the Cenozoic era facilitated the ecological success of diatoms in marine ecosystems and plunged coccolithophores into a long-term decline.

The biogeochemical consequences of this ecological switch were dramatic. The removal of C from Earth's surface reservoirs is constrained by the kinetics of chemical weathering reactions as it controls the burial of organic C, via input of

dissolved nutrients to the ocean basins, and $CaCO_3$, via alkalinity flux. The degree of coupling of organic C and $CaCO_3$ burial fluxes depends on the extent to which organic productivity is dominated by calcifying organisms (Bartley and Kah, 2004). A tight biological coupling becomes effective in the planktonic realm with the ecological success of planktonic calcifiers during the Mesozoic (Ridgwell, 2005). However, subsequent rearrangements of marine C cycling, triggered by competitive interactions between coccolithophores and diatoms, decreased the biological coupling of organic and inorganic

C burial fluxes, increasing the role of organic productivity in the removal of C from the atmosphere/ocean compartments. Our results illustrate how the interplay between continental geology, climate and ocean circulation influenced the evolution of marine primary producers and global biogeochemical cycles. The Cenozoic rise of diatoms in marine environments and the ensuing decline of coccolithophores accelerated the ocean's biologically-driven C sequestration (Cermeño, 2016).

Our results have implications for the design of ocean fertilization strategies aimed at enhancing the efficiency of the

ocean's biological pump and removing $CO_2$ from the atmosphere. Ultimately, the viability of ocean fertilization and its continuation over time spans long enough to accomplish targets depend on i) the extent to which nutrients are mobilized from geological reservoirs into marine environments (otherwise either N, P, Fe or Si would eventually become limiting), and ii) the fraction of captured C by photosynthesis that eventually is sequestered in the deep ocean and sediments (Smetacek et al., 2012). The former controls the oceanic inventory of inorganic nutrients and hence the capacity of the ocean to sequester

atmospheric $CO_2$ through biological processes (Falkowski et al., 1998). The latter dictates the efficiency of the C sequestration tool, which is largely dependent on the contribution of diatoms to export production and burial (Smetacek, 1999;Cermeño et al., 2008). It is well known that selective silica sequestration in the Southern Ocean limits the development of diatom blooms elsewhere and therefore the biologically-driven C sequestration potential of the entire ocean (Assmy et al., 2013). Thus, our results suggest that ocean fertilization initiatives would be more effective if they were integrated within the



framework of geo-engineering programs aimed at enhancing silicate weathering (Köhler et al., 2010;Schuiling and Krijgsman, 2006).

### Acknowledgements

P.C. was supported by a Ramón y Cajal contract from the Spanish Government. This work was supported by grant CTM2014-54926-R from the Spanish Ministry of Economy and Competitiveness.

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



**Figure legends:**

**Figure 1:** Conceptual scheme showing the effects of weathering fluxes and upper ocean turbulence on the ecological selection of coccolithophores and diatoms. (a) Changes in the coupling of P and Si weathering fluxes through time provide a basis to link the resource ratio theory of Tilman to the ecological selection of coccolithophores and diatoms. (b) The effect of turbulence on the dynamics of nutrient supply links the dynamical nutrient supply theory of Margalef to the ecological selection of coccolithophores and diatoms.

**Figure 2:** Map showing the geographic location of sediment samples from which P burial rate and lithium isotope data used in this study were obtained.

**Figure 3:** Continental weathering fluxes through time. (a) P weathering flux over the last 40 My based on the analysis of the sedimentary record (black dashed line) and model simulations (colour lines). (b) As in (a) but for Si weathering flux. The z-options represent different model configurations based on distinct combinations of the min/max Damköhler coefficient number (in meters per year): $5 \cdot 10^{-5}$ - 0.7 (*z1012*), 0.001 - 0.2 (*z1013*), 0.001 - 0.3 (*z1014*), 0.002 - 0.2 (*z1019*), 0.014 - 0.089 (*z1022*), 0.012 - 0.097 (*z1023*). Large Damköhler coefficients are indicative of steep terrains and high weathering fluxes characteristic of collisional environments.

**Figure 4:** (a) Si/P weathering flux ratio based on the sedimentary record (thin line) and model simulations (thick line). Dotted lines are the error bars associated with two different estimates of P weathering flux in modern rivers. (b) Coccolithophore and diatom SCOR (solid and dashed lines, respectively) [(Hannisdal et al. (2012) and Cermeño et al. (2015)]. (c) The diatom to coccolithophore SCOR ratio showing the rise of marine diatoms to ecological prominence.

25

30

35





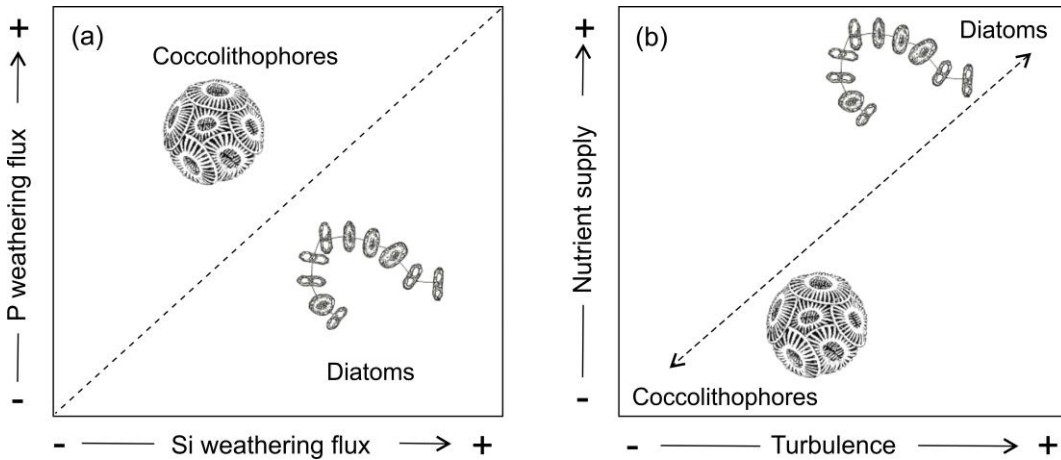

**Figure 1:** Conceptual scheme showing the effects of weathering fluxes and upper ocean turbulence on the ecological selection of coccolithophores and diatoms. (a) Changes in the coupling of P and Si weathering fluxes through time provide a basis to link the resource ratio theory of Tilman to the ecological selection of coccolithophores and diatoms. (b) The effect of turbulence on the dynamics of nutrient supply links the dynamical nutrient supply theory of Margalef to the ecological selection of coccolithophores and diatoms.



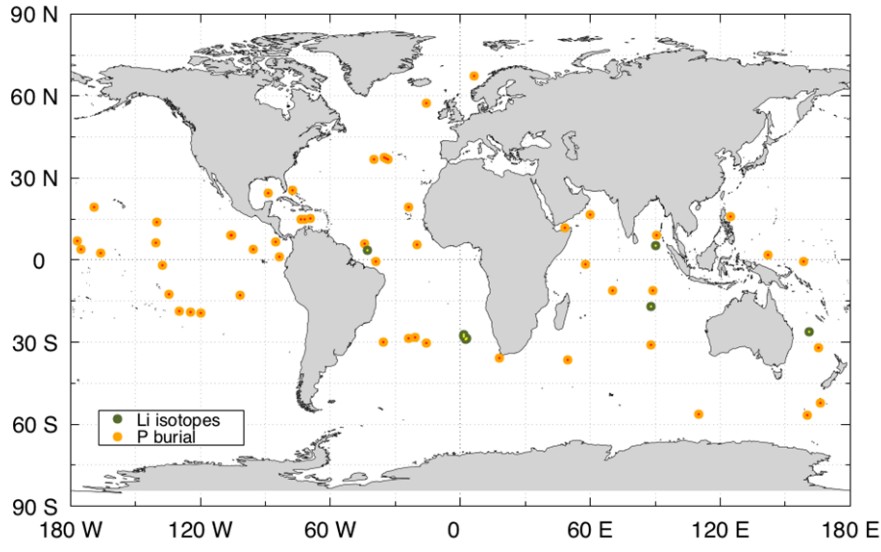

**Figure 2:** Map showing the geographic location of sediment samples from which P burial rate and lithium isotope data used in this study were obtained.

20



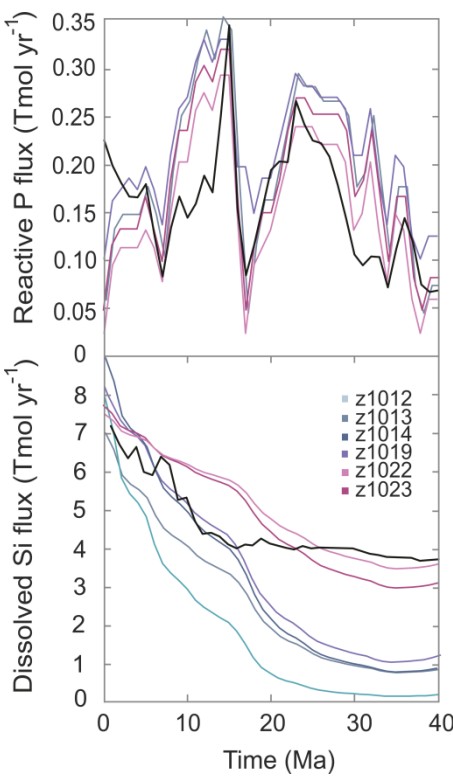

**Figure 3:** Continental weathering fluxes through time. (a) P weathering flux over the last 40 My based on the analysis of the sedimentary record (black dashed line) and model simulations (colour lines). (b) As in (a) but for the Si weathering flux. The z-options represent different model configurations based on distinct combinations of the min/max Damköhler coefficient number (in meters per year): $5 \cdot 10^{-5}$ - 0.7 (*z1012*), 0.001 - 0.2 (*z1013*), 0.001 - 0.3 (*z1014*), 0.002 - 0.2 (*z1019*), 0.014 - 0.089 (*z1022*), 0.012 - 0.097 (*z1023*). High Damköhler coefficents are indicative of steep terrains and high weathering fluxes characteristic of collisional environments.




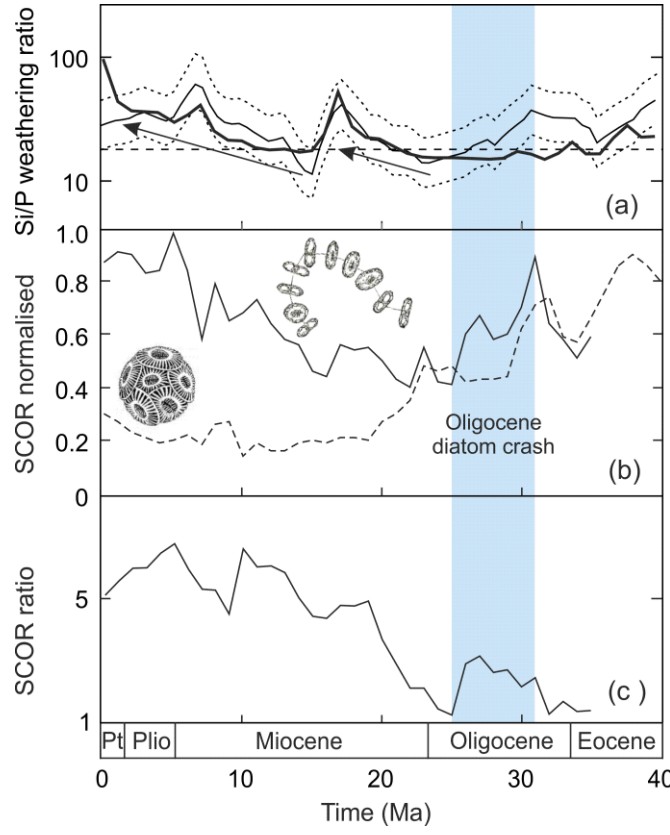

**Figure 4:** (a) Si/P weathering flux ratio based on the sedimentary record (thin line) and model simulations (thick line). Dotted lines are the error bars associated with two different estimates of P weathering flux in modern rivers (see Methods).

5 (b) Coccolithophore and diatom SCOR (dashed and solid lines, respectively) [(Hannisdal et al. (2012) and Cermeño et al. (2015)]. (c) The diatom to coccolithophore SCOR ratio showing the rise of marine diatoms to ecological prominence.