# Peer review of "Changes in the Si/P weathering ratio and their effect on the selection of coccolithophores and diatoms."

_Biogeosciences, 2017_

## Referee Comment (RC1) · Anonymous Referee #1 · 9 Mar 2017

Review summary: This study by Garcia-Bernal and colleagues presents some model- and proxy-based arguments for past changes in Si and P nutrient supply into the ocean, followed by a comparison to metrics aimed at measuring the dominance of coccolithophores and diatom phytoplankton. From that the authors conclude that Si:P nutrient supply ratio to the ocean was an important factor contributing to phytoplankton ecosystem evolution, but the alternative hypothesis of increasing turbulence was somehow at play too. The study concludes with a long discussion of tectonic, climatic and biological processes that may have caused changes in Si and P supply to the ocean, their knock-on effects on biological carbon sequestration, and the efficacy of geo-engineered weathering and ocean fertilization for drawdown of anthropogenic car-

bon. Overall, I find that the evidence base is not well enough explained/documented, the comparison between nutrient forcing and ecological response not convincing, and the discussion ignores some critical weaknesses in data and theory. Therefore, I am pessimistic about the prospect of this discussion paper.

Specific comments:

Hypotheses: The study offers two initial hypotheses for processes that may explain the relative rise of diatoms over coccolithophores (see Fig.1): increase in weathering Si:P supply to the ocean favoring silicifying organisms, or alternatively global deepening of the mixed layer favoring organisms with high specific and/or population growth rates. After presenting evidence that may or may not support increases in Si:P of nutrient supply the authors conclude (p.8, lines 32-33): "the distinct fortunes of coccolithophores and diatoms during the Cenozoic era cannot be attributed exclusively to changing weathering fluxes and nutrient ratios." Why not? So, what should I be taking away from this study?

Observational evidence for environmental forcing: Four pieces of observational evidence are used without appropriate discussion: (1) the Follmi (1995) dataset for P burial is used without discussion of the mechanisms of P diagenesis and burial and without mention that shelf/slope sediments account for ∼75% of P burial (see e.g. Baturin 2007) without being well represented in the Follmi dataset; (2) the lithium isotope record is used as a direct (and linear) proxy of silicate weathering although most colleagues would probably argue that it records secondary mineral formation and the congruency of weathering instead of the primary weathering reaction progress (see e.g. the original paper by Misra and Frolich, 2012); (3) SCOR is (p.4, lines 4-5) "a proxy of plankton functional group dominance" and no discussion is offered why this concept of "dominance" can be equated to relative biogeochemical importance (what is the SCOR for the most productive phytoplankton group, unicellular cyanobacteria?); (4) evidence for long-term increase in the pole-to-equator temperature gradient have important implications for deep water formation, the oceans overall density structure and meridional

overturning but it cannot be simply related to seasonal mixed layer depth and mixed layer light conditions as is done here.

Modeling: The model setup is insufficiently described. In particular the parameterization of the "weathering flux-uplift relationship (p5 line 19 to p6 line 3) is not transparent. What uplift record is used to force silica weathering? Likewise, it would be important to see what forcing gives rise to the simulated P weathering flux. I suspect simulated $CO_2$ could be a good criterion to discard unrealistic model scenarios, why is it not shown? How does the model Si weathering vary when using the default $CO_2$ dependent weathering scaling? As it is currently the model is a black box, the forcing is unknown, the output is incomplete and discussion is lacking.

Model-data and data-data comparison: Based on visual comparison the agreement between simulated and proxy-derived P and Si weathering fluxes is judged "remarkably coincident" and "satisfactory" (p 6 line 11,13). For P flux I find that the remarkable agreement cannot be coincidence, and for Si flux the poor match is unsatisfactory (model looks to follow strontium isotopes rather than lithium isotopes, the latter being used as the observational proxy). No objective analysis or relevant discussion is offered. Similarly, the authors find patterns in diatom and coccolithophores SCOR to be "consistent with changes in the Si/P weathering ratio (Fig. 4)" (p6 line 28-29). How so? I would have thought the various records are uncorrelated by any significance standard.

Discussion: The authors should seek to clarify the motivation for their discussion so as to avoid the sense that it aims to make ad hoc attribution of proposed changes in nutrient weathering fluxes to various tectonic, climate and environmental changes over the course of the Cenozoic. As one example, using denudation related to Himalayan orogeny as the core explanation for increases in silicate weathering after 20 Myrs even though that timing lags 15 Myrs behind seawater strontium isotope changes related to the same tectonic event is not helpful without detailed discussion of the discussion. Other aspects of the final section — such as the effect of the rise of diatoms

on the biological pump and atmospheric CO2 or the suggestion of geoengineering silica fertilization of the ocean to sequester anthropogenic carbon — are not sufficiently developed.
* * *

---

## Referee Comment (RC2) · G. F. de Souza (Referee) · 21 Mar 2017

In their manuscript, Dr. García Bernal and co-workers present an assessment of the interesting hypothesis that variation in the Si:P ratio of weathering fluxes to the ocean (together with changes in upper-ocean turbulence) might explain changes in the relative importance of marine coccolithophorids and diatoms observed in the sedimentary record over the last ∼40 million years. To do this, they reconstruct Si and P weathering rates from observations as well using an Earth system model, and compare the changes in the reconstructed Si:P ratio to changes in a metric for plankton functional group dominance. The authors state that this analysis suggests that the Si:P ratio (together with putative changes in upper-ocean turbulence) can explain the ecological

success of diatoms relative to coccolithophorids over the timeframe analysed. However, I cannot find any convincing evidence in the manuscript to back up this claim. This is mainly because the main results of the authors' work (i.e. a reconstruction of the Si:P of weathering flux and a reconstruction of phytoplankton dominance) are compared in a weak and non-quantitative fashion, simply by asserting similarity between the timeseries presented in Figure 4. To my eye, these timeseries do not show a strong similarity, and it would take a much more careful and rigorous analysis to convince me. In light of this, I cannot recommend this manuscript for publication in Biogeosciences in its current form. I provide some detailed comments below.

1. Model description:

I find the description of the model in Section 2.2. lacking in detail. On the one hand, the authors state that they apply a published model (COPSE; Bergman et al., 2006). On the other hand, they entirely alter the weathering flux-uplift relationship of that model, for both Si and P. Such an alteration may of course be justifiable, but requires much better documentation than the three lines devoted to the authors' approach at the beginning of page 6. Also, with respect to the oceanic P cycle in COPSE, it would be good if the authors could provide justification for the high C:P ratio of burial used by the model, rather than stating that an alternative parameterization (which is not used in this study, as far as I can tell) is available (P4, L30-32).

On a slightly different point, can the authors comment on what changes in the model lead to the massive decrease in P weathering flux and increase in Si flux at around 18Ma? Is this the effect of some external forcing to the model, and is it entirely independent of the P burial rate record from Föllmi (1995)?

2. Discussion of main results:

I think it is telling that the manuscript's Discussion section does not refer to the results of this study, but rather to general ideas about changes that might have occurred to weathering fluxes in the last 40 Ma. What this manuscript is missing is a convincing,

detailed analysis of its own results, beyond the assertion that the timeseries in Fig. 4 are consistent with each other. On the face of it I do not see any evidence for a close linkage between the records presented. Given this rather tenuous similarity, I would need some careful analysis before I could be convinced that they are at all related, but the authors only provide a qualitative descriptive comparison.

In my opinion, the authors need to address a few questions: - Are there likely threshold values in the Si:P of the ocean inventories (or the weathering flux) that might lead to non-linear coupling between this variable and relative phytoplankton dominance? - Could a simple analytical framework (e.g. a box model) be used to represent such thresholds/non-linearities and actually tie the records together and make them comparable in a slightly more quantitative way?

Additionally, the authors should spend some more time making sure that the records they present are understandable to the reader. Currently, the text does not clearly state what the important plankton metric is: is it the SCOR ratio or the normalised SCOR value? What different information can we get from these two? Currently, the two would seem to contradict each other in some cases (such as the relative dominance of diatoms during the putative "Oligocene diatom crash").

Minor comments:

P2, L13: The effect of inorganic:organic C rain ratio on atmospheric pCO2 goes back to Archer and Maier-Reimer (1994), and this work should be cited here rather than Cermeño et al. (2008).

P3, L3: The forces driving upper-ocean turbulence are not explained clearly. I assume that the authors mean the atmospheric temperature gradient between the equator and the poles and its effect on wind-driven mixing, but this should be clearly laid out for the reader.

P6, L20-23: I see neither the peak in Si:P of weathering flux at the E/O transition,

nor a contemporaneous peak in diatom SCOR values, in contrast with the authors' description.

Fig. 1: I have a problem with panel b. Within the context of ocean-internal nutrient cycling, I would argue that an increase in upper-ocean turbulence results in an increase in nutrient supply (through increased vertical mixing), and thus the two axes of this panel fall together. Regardless of this, to be entirely conceptually correct, the diagram should show the cartoons for coccolithophores and diatoms on the line, rather than on either side of it.

Fig. 4: Are the error bars in panel a associated with the range of 0.1-0.3 Tmol P/yr mentioned in the main text? If so, it would be good to mention this explicitly in the caption.

---

## Author Comment (AC1) · 8 May 2017

We thank the reviewer for his/her helpful comments

R.C.: Reviewer comment

A.R.: Authors response

R.C: Hypotheses: The study offers two initial hypotheses for processes that may explain the relative rise of diatoms over coccolithophores (see Fig.1): increase in weathering Si:P supply to the ocean favouring silicifying organisms, or alternatively global deepening of the mixed layer favouring organisms with high specific and/or population growth rates. After presenting evidence that may or may not support increases in Si:P

of nutrient supply the authors conclude (p.8, lines 32-33): "the distinct fortunes of coc-colithophores and diatoms during the Cenozoic era cannot be attributed exclusively to changing weathering fluxes and nutrient ratios." Why not? So, what should I be taking away from this study?

A.R: The ecological theories of nutrient supply ratio and nutrient supply dynamics are strongly supported by a large body of theoretical and experimental work. These theories are not mutually exclusive, e.g. an increase in the Si/P supply ratio is a necessary but insufficient condition for the rise of diatoms, which also require intermittent pulses of nutrient supply to outcompete other phytoplankton such as coccolithophores. On ge-ological time scales, nutrient supply ratios are controlled by weathering ratios, whereas nutrient supply dynamics are dependent on wind-driven upper ocean turbulence (and the frequency of nutrient pulses to the upper mixed layer). Both features are essential requirements for the ecological success of diatoms. Previous work has focused on ocean turbulence (Tozzi et al., 2004; Falkowski and Oliver, 2007). However, the ef-fect of weathering ratios remains underexplored. We explicitly clarify these apparently confusing aspects in the new version of the manuscript.

R.C: Observational evidence for environmental forcing: Four pieces of observational evidence are used without appropriate discussion: (1) the Follmi (1995) dataset for P burial is used without discussion of the mechanisms of P diagenesis and burial and without mention that shelf/slope sediments account for 75% of P burial (see e.g. Ba-turin 2007) without being well represented in the Follmi dataset.

A.R: We agree with the reviewer that the Follmi dataset for P burial is biased towards abyssal sediments whereas much of the marine P cycle takes place along continental margins. We comment on this issue in a new version of the manuscript and cite relevant literature on the topic. Regardless of this limitation, our modelling approach, aimed at supporting estimates derived from sedimentary proxies, did a reasonably good job.

R.C: (2) the lithium isotope record is used as a direct (and linear) proxy of silicate

weathering although most colleagues would probably argue that it records secondary mineral formation and the congruency of weathering instead of the primary weathering reaction progress (see e.g. the original paper by Misra and Froelich, 2012);

A.R: We agree with the reviewer that the lithium isotope record does not provide a direct estimate of weathering rates but an indication of whether the dominant regime was weathering-limited or supply-limited (i.e. the weathering style). Weathering-limited regimes dominate in tectonically-active regions where intense physical erosion maintains fresh rock surfaces continuously exposed to chemical degradation. Conversely, supply-limited regimes dominate in flat terrains where thick soils prevent mineral surfaces from contact with the atmosphere and hence weathering rates are low. We did not use the lithium isotope record in strict proportionality to estimate weathering rates, but as a proxy for the dominant weathering regime. Our strategy was to assume a two-fold increase in silicate weathering over the last 40 million years following (Li and Elderfield, 2013)(computed from inverse models) and then, based on the Li isotope record of weathering styles, the curve of dissolved Si flux to the ocean basins was computed accordingly. Our calculation assumes that the particulate to dissolved flux ratio in the past was 1:4 whereas currently this ratio takes a value of 4:1. A thorough explanation is given in (Cermeño et al., 2015) (including a *.xls file with pertinent calculations in Supplementary information).

In the new version of the manuscript we include a more thorough explanation in methods on assumptions considered to obtain estimates of dissolved Si fluxes through time.

R.C: (3) SCOR is (p.4, lines 4-5) "a proxy of plankton functional group dominance" and no discussion is offered why this concept of "dominance" can be equated to relative biogeochemical importance (what is the SCOR for the most productive phytoplankton group, unicellular cyanobacteria?);

A.R: The SCOR index has been used to quantify the dominance of plankton functional groups such as coccolithophores, diatoms, foraminifera and radiolarians (Liow et al.,

2010; Hannisdal et al., 2012; Cermeño et al., 2015). Given that the microfossil record is largely limited to data of species presence/absence, estimates of dominance are commonly based on taxonomic richness (rather than abundance). However, taxonomic richness is not necessarily indicative of biogeochemical significance. The PaleoBiology database has global coverage, which allows us to compute the extent of geographic distribution (SCOR). The SCOR is thus a more realistic measure of biogeochemical significance than diversity. Unicellular cyanobacteria are ubiquitous and responsible for a large fraction of marine primary production in open ocean systems. Yet the lack of fossil record (not considering biochemical markers) and our inability to taxonomically resolve the group without the aid of molecular techniques prevent us from computing their SCOR.

R.C: (4) evidence for long-term increase in the pole-to-equator temperature gradient have important implications for deep water formation, the oceans overall density structure and meridional overturning but it cannot be simply related to seasonal mixed layer depth and mixed layer light conditions as is done here.

A.R: The pole-to-equator temperature gradient influences the vertical density structure of the oceans but also atmospheric circulation and wind patterns. The long-term increase in the pole-to-equator temperature gradient is expected to intensify atmospheric circulation patterns and wind stress, which enhanced upper ocean turbulence and upwelling along continental margins and equatorial divergences.

R.C: Modeling: The model setup is insufficiently described. In particular the parameterization of the "weathering flux-uplift relationship (p5 line 19 to p6 line 3) is not transparent. What uplift record is used to force silica weathering? Likewise, it would be important to see what forcing gives rise to the simulated P weathering flux. I suspect simulated $CO_2$ could be a good criterion to discard unrealistic model scenarios, why is it not shown? How does the model Si weathering vary when using the default $CO_2$ dependent weathering scaling? As it is currently the model is a black box, the forcing is unknown, the output is incomplete and discussion is lacking.

A.R: We used the uplift forcing provided as "default input" in COPSE version 5, which is based on the strontium isotope data from the Lowess fit (McArthur et al., 2001). In our modified version of COPSE, weathering is independent of atmospheric $CO_2$ concentration. As it stands, weathering depends exclusively on the new uplift-weathering parameterization derived from (Maher and Chamberlain, 2014). We are undergoing new simulations to test both the combined effect of uplift and volcanic degassing on weathering fluxes.

Si weathering from COPSE's default configuration decreases significantly towards the present primarily associated with a reduction in volcanic degassing. In the original model configuration, Si weathering is strongly dependent on volcanic degassing despite the latter being poorly constrained by data. Strontium isotopes, in contrast, suggest an increase in continental weathering and erosion coincident with the uplift of the Himalayas. This apparent controversy is further discussed in the new version of the manuscript as it seems to be a critical aspect of the model, at least for the last 40 million years.

R.C: Model-data and data-data comparison: Based on visual comparison the agreement between simulated and proxy-derived P and Si weathering fluxes is judged "remarkably coincident" and "satisfactory" (p 6 line 11,13). For P flux I find that the remarkable agreement cannot be coincidence, and for Si flux the poor match is unsatisfactory (model looks to follow strontium isotopes rather than lithium isotopes, the latter being used as the observational proxy). No objective analysis or relevant discussion is offered. Similarly, the authors find patterns in diatom and coccolithophores SCOR to be "consistent with changes in the Si/P weathering ratio (Fig. 4)" (p6 line 28-29). How so? I would have thought the various records are uncorrelated by any significance standard.

A.R: Our study attempts to put together estimates of P and Si flux to the ocean basins in order to explore the effect of the weathering flux ratio on the ecological success of marine diatoms. We previously realized that this specific issue has received little attention. Our strategy was to implement an Earth system biogeochemical model by adding a new parameterization for uplift-weathering following (Maher and Chamberlain, 2014) and then compare the simulation outputs with sedimentary proxy data. The reviewer notes that our Si weathering flux curve seems to follow strontium isotopes rather than lithium. We find no evidence supporting this claim – whereas the strontium isotope curve begins to increase remarkably roughly 40 million years ago, our Si weathering flux curve highlights a major change 20 million years ago, presumably associated with the thrusting of the Himalayas. The lack of a step-like curve as depicted by the lithium isotope record has to do with the fact that the pattern becomes smoothed as a result of the assumptions considered in our calculations [i.e., the particulate to dissolved flux ratio in the past was 1:4 whereas currently this ratio takes a value of 4:1]. Though there are differences between our estimates and those produced by the model (after some implementation and tuning of parameters), overall, the patterns were very much similar.

We also agree with the reviewer that the presumed relationship between the Si/P weathering flux ratios and the SCOR ratio is not straightforward. In future work we will need to carry out further studies at a higher temporal resolution to investigate such a linkage and its interrelationship with other factors; notwithstanding the Earth is a complex system with many independent forcing mechanisms acting simultaneously, often, in non-linear fashion. Experimental ecologists use to isolate factors and look at their potential effects on organisms, populations and communities. The geological record provides an averaged signature of multiple causes and effects, which clouds specific cause-and-effect relationships and precludes from a comprehensive understanding of specific mechanisms at play. Rather we report on a previously unexplored Si/P weathering ratio and discuss some potential interpretations of the linkage between continental weathering, ocean hydrodynamics and phytoplankton ecology. We think that the analysis add new and interesting information to compose a more detailed picture of the mechanisms that rose diatoms to ecological prominence.

R.C: Discussion: The authors should seek to clarify the motivation for their discussion so as to avoid the sense that it aims to make ad hoc attribution of proposed changes in nutrient weathering fluxes to various tectonic, climate and environmental changes over the course of the Cenozoic. As one example, using denudation related to Himalayan orogeny as the core explanation for increases in silicate weathering after 20 Myrs even though that timing lags 15 Myrs behind seawater strontium isotope changes related to the same tectonic event is not helpful without detailed discussion of the discussion. Other aspects of the final section — such as the effect of the rise of diatoms on the biological pump and atmospheric CO2 or the suggestion of geoengineering silica fertilization of the ocean to sequester anthropogenic carbon — are not sufficiently developed.

A.R: In a new version of the manuscript we include a more detailed discussion on the mechanisms underlying the patterns of Si and P weathering fluxes to the ocean basins and their influence on the ecological success of diatoms. Additionally, we elaborate on additional aspects concerning potential geoengineering implications of our results.

References cited:

Cermeño, P., Falkowski, P. G., Romero, O. E., Schaller, M. F., and Vallina, S. M.: Continental erosion and the Cenozoic rise of marine diatoms, Proceedings of the National Academy of Sciences, 112, 4239-4244, 10.1073/pnas.1412883112, 2015. Falkowski, P. G., and Oliver, M. J.: Mix and match: how climate selects phytoplankton, Nat Rev Micro, 5, 813-819, 2007. Hannisdal, B., Henderiks, J., and Liow, L. H.: Long-term evolutionary and ecological responses of calcifying phytoplankton to changes in atmospheric CO2, Global Change Biology, 18, 3504-3516, 10.1111/gcb.12007, 2012. Li, G., and Elderfield, H.: Evolution of carbon cycle over the past 100 million years, Geochimica et Cosmochimica Acta, 103, 11-25, http://dx.doi.org/10.1016/j.gca.2012.10.014, 2013. Liow, L. H., Skaug, H. J., Ergon, T., and Schweder, T.: Global occurrence trajectories of microfossils: environmental volatility and the rise and fall of individual species, Paleobiology, 36, 224-252, 10.1666/08080.1, 2010. Maher, K., and Chamberlain, C.:

Hydrologic regulation of chemical weathering and the geologic carbon cycle, Science, 343, 1502-1504, 2014. McArthur, J. M., Howarth, R. J., and Bailey, T. R.: Strontium Isotope Stratigraphy: LOWESS Version 3: Best Fit to the Marine Sr Isotope Curve for 0–509 Ma and Accompanying Look up Table for Deriving Numerical Age, The Journal of Geology, 109, 155-170, doi:10.1086/319243, 2001. Tozzi, S., Schofield, O., and Falkowski, P. G.: Historical climate change and ocean turbulence as selective agents for two key phytoplankton functional groups, Marine Ecology Progress Series, 274, 123-132, 2004.

---

## Author Comment (AC2) · 8 May 2017

We thank Dr. de Souza for his constructive criticisms and helpful comments.

R.C.: Reviewer comment

A.R.: Authors response

R.C: The authors state that this analysis suggests that the Si:P ratio (together with putative changes in upper-ocean turbulence) can explain the ecological success of diatoms relative to coccolithophorids over the timeframe analysed. However, I cannot find any convincing evidence in the manuscript to back up this claim. This is mainly because the main results of the authors' work (i.e. a reconstruction of the Si:P of

weathering flux and a reconstruction of phytoplankton dominance) are compared in a weak and non-quantitative fashion, simply by asserting similarity between the time series presented in Figure 4. To my eye, these time series do not show a strong similarity, and it would take a much more careful and rigorous analysis to convince me.

A.R: This issue has been expressed by both reviewers. Our analysis was not intended to demonstrate a strong coupling between nutrient weathering ratios and phytoplankton ecology. As stated in our response to Reviewer 1, the Earth is a complex system with many independent forcing mechanisms acting simultaneously, and often in a non-linear fashion. The geological record provides an averaged signature of a myriad of factors, which preclude us from identifying satisfactorily specific cause-and-effect relationships. Thus, though our analysis is purely descriptive (i.e., we were motivated by the realization that there was a conspicuous lack of knowledge concerning the linkage between weathering flux ratios and plankton ecology over geologic time), yet, some features emerge that may help to improve our understanding of the linkage between continental weathering ratios, ocean hydrodynamics and phytoplankton ecology. For instance, we find that throughout the Cenozoic the Si/P weathering flux ratio is commonly well above the classical Redfield. Additionally, some consistent patterns in the Si/P weathering ratio from both sediment data and model simulations such as the peaks observed in the mid- and late-Miocene deserve further scrutiny. We think that the analysis adds new and interesting information to compose a more detailed picture of the mechanisms that rose diatoms to ecological prominence. Our analysis may serve as a starting point to undertake some of the challenging questions raised by the reviewer below (i.e. Are there likely threshold values in the Si:P of the ocean inventories (or the weathering flux) that might lead to non-linear coupling between this variable and relative phytoplankton dominance?)

R.C: Model description: I find the description of the model in Section 2.2. lacking in detail. On the one hand, the authors state that they apply a published model (COPSE; Bergman et al., 2006). On the other hand, they entirely alter the weathering flux-uplift

relationship of that model, for both Si and P. Such an alteration may of course be justifiable, but requires much better documentation than the three lines devoted to the authors' approach.

A.R: We provide a thorough description of the changes included and the extent to which these changes altered the outputs.

Si weathering from COPSE's default configuration decreases significantly towards the present primarily associated with a reduction in volcanic degassing. In the original model configuration, Si weathering is strongly dependent on volcanic degassing despite the latter being poorly constrained by data. Strontium isotopes, in contrast, suggest an increase in continental weathering and erosion coincident with the uplift of the Himalayas. This apparent controversy is further discussed in the new version of the manuscript as it seems to be a critical aspect of the model, at least for the last 40 million years. We elaborate on a more detailed justification of the new weathering-uplift parameterization.

R.C: Also, with respect to the oceanic P cycle in COPSE, it would be good if the authors could provide justification for the high C:P ratio of burial used by the model, rather than stating that an alternative parameterization (which is not used in this study, as far as I can tell) is available (P4, L30-32).

A.R: We have used the parameterization set by default in the initial model configuration. We intended to alter external forcing mechanisms associated with uplift and weathering yet preserving other critical aspects of the original model.

R.C: On a slightly different point, can the authors comment on what changes in the model lead to the massive decrease in P weathering flux and increase in Si flux at around 18Ma? Is this the effect of some external forcing to the model, and is it entirely independent of the P burial rate record from Föllmi (1995)?

A.R: Yes, the model analysis is entirely independent of the Follmi record. Si flux is primarily driven by uplift and silicate rock weatherability. On the other hand, P weathering depends on weathering of silicates and carbonates, and the concurrent oxidation of sedimentary organic matter by assuming a relative proportion of P in each rock type.

R.C: Discussion of main results: I think it is telling that the manuscript's Discussion section does not refer to the results of this study, but rather to general ideas about changes that might have occurred to weathering fluxes in the last 40 Ma. What this manuscript is missing is a convincing detailed analysis of its own results, beyond the assertion that the time series in Fig. 4 are consistent with each other. On the face of it I do not see any evidence for a close linkage between the records presented. Given this rather tenuous similarity, I would need some careful analysis before I could be convinced that they are at all related, but the authors only provide a qualitative descriptive comparison.

A.R: As stated before, we cannot establish a straight relationship between weathering flux ratios and the temporal distribution of plankton groups such as diatoms and coc- colithophores. This limitation lies in the fact that other potential mechanisms are acting simultaneously and the analysis of the sedimentary/fossil record does not allow us to easily discern patterns from mechanisms.

R.C: In my opinion, the authors need to address a few questions: - Are there likely threshold values in the Si:P of the ocean inventories (or the weathering flux) that might lead to non-linear coupling between this variable and relative phytoplankton dom- inance? - Could a simple analytical framework (e.g. a box model) be used to represent such thresholds/non-linearities and actually tie the records together and make them comparable in a slightly more quantitative way?

A.R: These are very insightful ideas. Ecological studies suggest that diatoms dominate phytoplankton communities as long as Si exceeds 2 micromolar and other nutrients are not limiting. Further studies have shown that the Si/P nutrient supply ratio also exerts a strong control on diatom dominance relative to other phytoplankton(Egge and

Aksnes, 1992; Egge, 1998). Upper ocean turbulence and Si weathering fluxes have been put forward to explain the ecological success of diatoms in marine environments over geological time scales (Falkowski and Oliver, 2007; Cermeño et al., 2015). Here we include the Si/P weathering ratio as another potential control on diatom ecological success. We recognize that our analysis is insufficient to resolve the questions raised by the reviewer, but they help to expand the picture by showing that the Si/P weathering flux ratio is well above Redfield throughout most of the second half of the Cenozoic, when upper ocean turbulence and increased Si supplies contributed to the expansion of diatoms in marine environments.

R.C: Additionally, the authors should spend some more time making sure that the records they present are understandable to the reader. Currently, the text does not clearly state what the important plankton metric is: is it the SCOR ratio or the normalised SCOR value? What different information can we get from these two? Currently, the two would seem to contradict each other in some cases (such as the relative dominance of diatoms during the putative "Oligocene diatom crash").

A.R: The SCOR index has been used to quantify the dominance of plankton functional groups such as coccolithophores, diatoms, foraminifera and radiolarians (e.g. Liow et al. 2011, Hannisdal et al. 2011, Cermeño et al. 2015). Given that the microfossil record is largely limited to data of species presence/absence, estimates of dominance are commonly based on taxonomic richness (rather than abundance). However, taxonomic richness is not necessarily indicative of biogeochemical significance. The PaleoBiology database has global coverage, which allows us to compute the extent of geographic distribution (SCOR). The SCOR is thus a more realistic measure of biogeochemical significance than diversity. Normalised SCOR values were used to adjust both diatom and coccolithophore SCOR into the same axis. On the other hand, we used the SCOR ratio to illustrate changes in the relative importance of each phytoplankton group through time. These distinct metrics are now explained thoroughly.

R.C: Minor comments: P2, L13: The effect of inorganic:organic C rain ratio on atmospheric pCO2 goes back to Archer and Maier-Reimer (1994), and this work should be cited here rather than Cermeño et al. (2008).

A.R: Done

R.C: P3, L3: The forces driving upper-ocean turbulence are not explained clearly. I assume that the authors mean the atmospheric temperature gradient between the equator and the poles and its effect on wind-driven mixing, but this should be clearly laid out for the reader.

A.R: Exactly, the amplification of temperature gradient between the equator and the poles intensified atmospheric circulation and wind-driven mixing. This is now explained in detail.

R.C: P6, L20-23: I see neither the peak in Si:P of weathering flux at the E/O transition nor a contemporaneous peak in diatom SCOR values, in contrast with the authors' description.

A.R: The Si/P weathering ratio from the sedimentary record increases across the Eocene/Oligocene. This is not observed however from the analysis of model outputs. This is commented in the new version of the manuscript. The diatom Oligocene crash is particularly visible when we look at the diatom SCOR in Fig 2b. Diatoms exhibit a conspicuous decrease during the Oligocene that is also observable for diversity estimates (Rabosky and Sorhannus, 2009). This is surprising because this diatom crash is preceded by a rapid expansion across the E/O transition. We suggest that this increase in diatom diversity and SCOR could be associated in part with the high Si/P nutrient ratios observed in the sedimentary record. This observation and the discrepancy in the E/O weathering ratio between the sedimentary record and model outputs are now clarified in the new version of the manuscript.

R.C: Fig. 1: I have a problem with panel b. Within the context of ocean-internal nutrient cycling, I would argue that an increase in upper-ocean turbulence results in an increase

in nutrient supply (through increased vertical mixing), and thus the two axes of this panel fall together. Regardless of this, to be entirely conceptually correct, the diagram should show the cartoons for coccolithophores and diatoms on the line, rather than on either side of it.

A.R: We have repositioned the cartoons accordingly.

R.C: Fig. 4: Are the error bars in panel a associated with the range of 0.1-0.3 Tmol P/yr mentioned in the main text? If so, it would be good to mention this explicitly in the caption.

A.R: Yes, the meaning of the error bars is now mentioned in the text.

References cited:

Cermeño, P., Falkowski, P. G., Romero, O. E., Schaller, M. F., and Vallina, S. M.: Continental erosion and the Cenozoic rise of marine diatoms, Proceedings of the National Academy of Sciences, 112, 4239-4244, 10.1073/pnas.1412883112, 2015.

Egge, J., and Aksnes, D.: Silicate as regulating nutrient in phytoplankton competition, Marine Ecology Progress Series, 83, 281-289, 1992.

Egge, J.: Are diatoms poor competitors at low phosphate concentrations?, Journal of Marine Systems, 16, 191-198, 1998.

Falkowski, P. G., and Oliver, M. J.: Mix and match: how climate selects phytoplankton, Nat Rev Micro, 5, 813-819, 2007.

Rabosky, D. L., and Sorhannus, U.: Diversity dynamics of marine planktonic diatoms across the Cenozoic, Nature, 457, 183-186, 2009.